# Two-electron transfer stabilized by excited-state aromatization

Jinseok Kim[1,6], Juwon Oh[1,6], Seongchul Park[2], Jose L. Zafra[3], Justin R. DeFrancisco[4], David Casanova[5]*, Manho Lim[2]*, John D. Tovar[4]*, Juan Casado [3]* & Dongho Kim[1]*

The scientific significance of excited-state aromaticity concerns with the elucidation of processes and properties in the excited states. Here, we focus on **TMTQ**, an oligomer composed of a central 1,6-methano[10]annulene and 5-dicyanomethyl-thiophene peripheries (acceptor-donor-acceptor system), and investigate a two-electron transfer process dominantly stabilized by an aromatization in the low-energy lying excited state. Our spectroscopic measurements quantitatively observe the shift of two π-electrons between donor and acceptors. It is revealed that this two-electron transfer process accompanies the excited-state aromatization, producing a Baird aromatic 8π core annulene in **TMTQ**. Biradical character on each terminal dicyanomethylene group of **TMTQ** allows a pseudo triplet-like configuration on the 8π core annulene with multiexcitonic nature, which stabilizes the energetically unfavorable two-charge separated state by the formation of Baird aromatic core annulene. This finding provides a comprehensive understanding of the role of excited-state aromaticity and insight to designing functional photoactive materials.

[1] Spectroscopy Laboratory for Functional π-electronic Systems and Department of Chemistry, Yonsei University, Seoul 03722, Korea. [2] Department of Chemistry and Chemistry Institute for Functional Materials, Pusan National University, Busan 46241, Korea. [3] Department of Physical Chemistry, University of Málaga, Andalucia-Tech, Campus de Teatinos s/n, 29071 Málaga, Spain. [4] Department of Chemistry, Johns Hopkins University, 3400 North Charles Street, Baltimore, MD 21218, USA. [5] Donostia, International Physics Center (DIPC) & IKERBASQUE - Basque Foundation for Science, Paseo Manuel de Lardizabal, 4, 20018 Donostia-San Sebastián, Euskadi, Spain. [6] These authors contributed equally: Jinseok Kim, Juwon Oh. *email: david.casanova@ehu.eus; mhlim@pusan.ac.kr; jtovar1@jhu.edu; casado@uma.es; dongho@yonsei.ac.kr

Aromaticity plays a key role in the understanding of molecular properties and chemical reactions because of special energetic landscape arising from cyclic π-conjugation[1]. Recently, the attention on aromaticity has shifted from the ground state to the excited states. In the same way that molecular properties are largely affected by the ground-state aromaticity, excited-state aromaticity can guide the rationalization of excited-state processes and photochemistry, and, ultimately, will provide the ad hoc design and development of novel functional materials[2].

For excited-state aromaticity, a reversal of ground-state aromaticity in the excited triplet state was proposed in 1972, known as Baird's rule[3]. Considering cyclic π-conjugated systems in the triplet state as an intermolecular interaction between two radicals, the interaction between two symmetric singly occupied molecular orbitals (SOMOs) in $[4n + 2]$π-electron systems leads to an energetic destabilization[2,4]. On the other hand, despite the lack of SOMO–SOMO interaction in $[4n]$π-electron systems, the interaction of a SOMO with energetically adjacent symmetric occupied and unoccupied molecular orbitals stabilizes the $[4n]$π cyclic conjugation interaction. This completely reverses the criteria for excited-state aromaticity (that is, aromaticity/antiaromaticity for $[4n]/[4n + 2]$π-electron systems in the triplet states) and has been intensively investigated on a theoretical basis[5–8]. This concept has also been experimentally supported by the evaluation of photochemical reactions where changes of aromaticity in the excited intermediate states serve as reaction driving forces[9–14].

This scenario enables us to open up a new scientific possibility dealing with the control of aromaticity in the excited states which allows us to anticipate the outcome of excited-state phenomena. In other words, understanding the role of aromaticity in the excited states enables the mechanistic elucidation of photoinduced electronic properties[1,2,15]. In this regard, shifts of π-electron density during the formation of photo-activated intermediate states, such as charge transfer/separation and multiple exciton/charge generation[16–20], can be driven by changes of aromaticity. Recent intensive efforts by optical and transient spectroscopic analyses have paved the way to unravel the effect of excited-state aromaticity[14,21–23]. Time-resolved optical spectroscopies enable us to monitor (anti)aromaticity-driven changes of electronic structure and molecular conformations, which will fundamentally help to understand the role and effect of aromaticity in the excited states.

For revealing the effect of excited-state aromaticity, a modulation of aromaticity in the excited state is prerequisite. Typically, molecular modification results in changes of both ground- and excited-state nature. In order to understand the aromaticity effects in the excited states, electronic effects that solely appear in the excited states without perturbing the ground-state nature are essential. Taking this into consideration, excited-state charge transfer (CT) processes, a shift of π-electron density between specific π-conjugated moieties in the excited states, can be a very effective approach to modulate the aromaticity only in the excited states.

Here, we focus on an π-conjugated oligomer (TMTQ; Fig. 1), which is composed of a central 1,6-methano[10]annulene (M10A) and 5-dicyanomethyl-thiophene (DT) peripheries in exo geometry (Supplementary Figs. 1 and 2)[24,25]. In TMTQ, the DT units serve as electron acceptors (A) due to the strong electron-withdrawing nature of the dicyanomethylene groups. On the other hand, the non-aromatic core annulene of TMTQ, due to its highly distorted and quinoidal structure, facilitates M10A to act as an electron donor (D). This A–D–A segmentation of TMTQ is an ideal configuration to promote excited-state intramolecular CT (iCT) processes[26,27]. Along the symmetric A–D–A geometry, it is expected that the iCT process gives rise to a shift of two

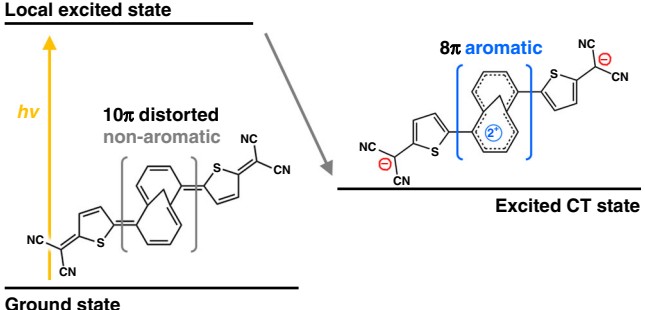

**Fig. 1** Charge-transfer-induced excited-state aromatization. The molecular structure of TMTQ and schematic illustration for an excited-state aromatization induced by intramolecular CT process

π-electrons from M10A to DT units. Based on Baird's rule, this can lead to an aromatization of the core annulene in M10A by a change of its 10π electrons in the ground state into 8π electrons in the CT state.

According to this, we demonstrate here the aromatization of TMTQ in the excited state induced by the iCT process via optical spectroscopic measurements and analyze the role of aromatization in the excited-state dynamics. In this study, we directly observed aromatization in the excited state induced by iCT process, suggesting that the control and modulation of photoinduced charge-separation mechanisms can be one of most effective approaches to control the excited-state aromaticity. Furthermore, this observation gives a comprehensive understanding of the effect of excited-state aromaticity, which will provide crucial insights into the application of excited-state aromaticity for the design of functional materials.

## Results

**Intramolecular CT in the excited state**. TMTQ showed an intense absorption in the range of 500–800 nm without any fluorescence regardless of solvent polarity (toluene, $CH_2Cl_2$, and $CH_3NO_2$; Fig. 2a). The non-fluorescent nature of TMTQ even under non-polar toluene condition suggests an effective iCT process[28].

Since the stabilization of the CT state by solvent polarity is expected to largely affect excited-state dynamics[29], we measured femtosecond transient absorption (fs-TA) spectra of TMTQ in toluene, $CH_2Cl_2$, and $CH_3NO_2$. The TA spectra of TMTQ in toluene, $CH_2Cl_2$, and $CH_3NO_2$ showed ground-state bleaching (GSB) bands from 550 to 800 nm and strong photoinduced absorption (PIA) bands in the NIR region (800 –1100 nm) in Fig. 2b–d. Despite the similar TA spectral features of TMTQ, a distinct difference was observed in the TA decay profiles upon the change of solvent (Supplementary Table 1, Supplementary Figs. 3 and 4). After the common fast decay of the TA signal (<40 fs), TMTQ in toluene showed subsequent double exponential decays (1.3 and 40 ps). With an increase in solvent polarity, the subsequent exponential decays of the TA signal become faster (1.0, 9.0, and 15 ps in $CH_2Cl_2$ and 0.8, 2.5, and 19 ps in $CH_3NO_2$). This decrease in the excited-state lifetime of TMTQ upon increasing the solvent polarity suggests the existence of an excited state with a charge-localized electronic structure, which is more stabilized under more polar solvent conditions and accelerates the relaxation processes in the excited state dynamics[29,30]. Therefore, in spite of the negligible difference in the steady-state absorption spectra of TMTQ upon the change of solvent polarity, the distinct dependence of TA decay profiles on the solvent polarity clearly indicates the iCT process only occurs in the excited state.

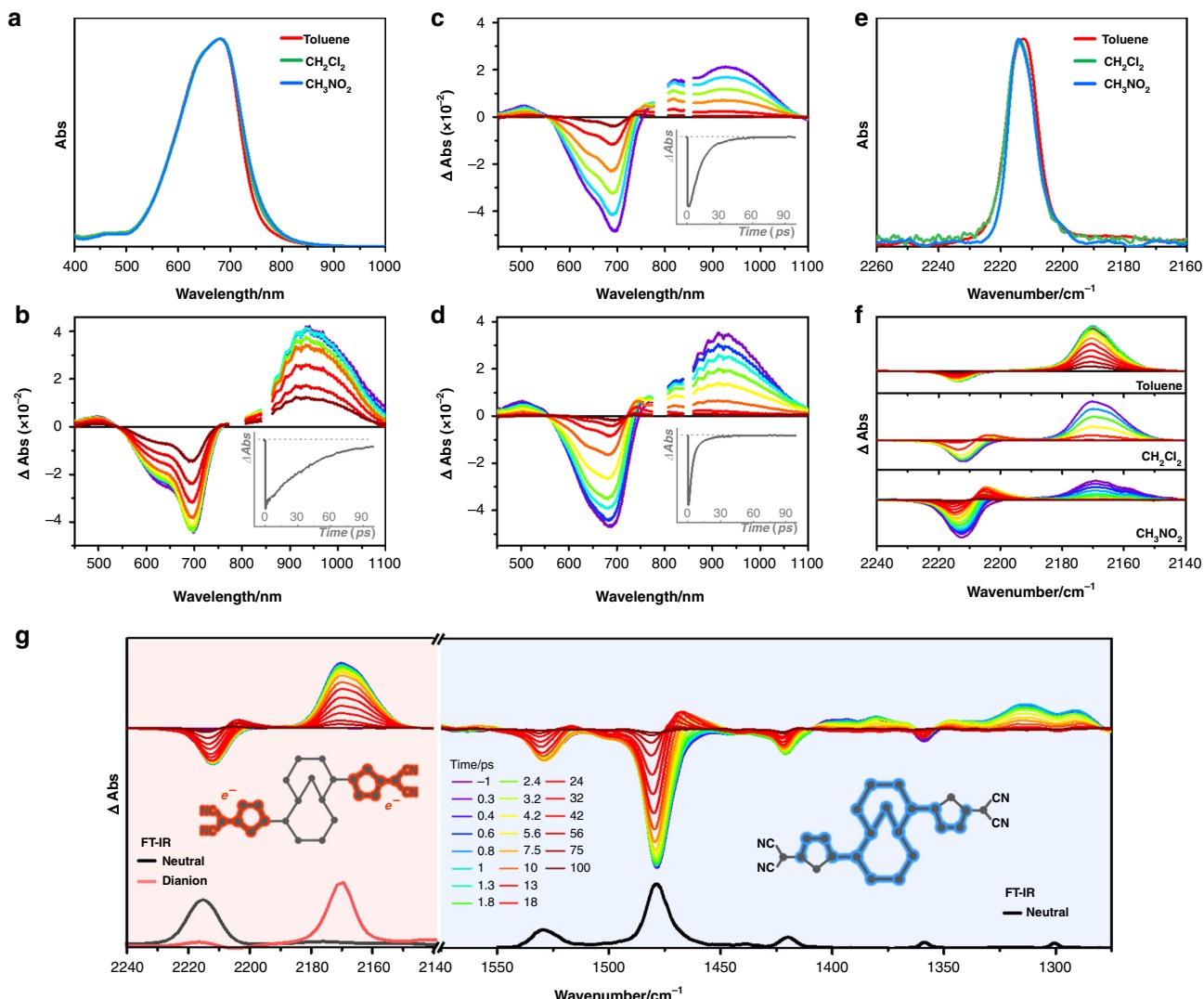

**Fig. 2** Electronic and vibrational spectra of TMTQ. **a–d** The steady-state (**a**) and transient absorption spectra of TMTQ in toluene (**b**), $CH_2Cl_2$ (**c**), and $CH_3NO_2$ (**d**). The inset plots are the decay profiles at 660 nm. **e**, **f** The FT-IR (**e**) and transient IR (**f**) spectra of TMTQ in toluene, $CH_2Cl_2$, and $CH_3NO_2$. The transient IR spectra were plotted within ~100 ps time window (from purple to dark red colored lines). **g** The transient IR spectra of TMTQ in $CH_2Cl_2$. The spectra in the regions of 2140–2240 and 1275–1575 $cm^{-1}$ are assigned to IR bands for C≡N and C=C stretching vibrations, respectively. The FT-IR spectra of neutral and electrochemically produced dianion TMTQ are inset

**Quantitative analysis of the excited-state CT.** The C≡N stretching mode of the dicyanomethylene group is well known as an effective marker for the quantitative analysis of electron density on the dicyano groups, where an increase (decrease) of electron density on the dicyanomethylene group leads to a red-shift (blue-shift) of the C≡N stretching band[31–35]. In this regard, we measured the IR spectra of **TMTQ** in the ground and excited states by Fourier-transform IR (FT-IR) and transient IR spectroscopies, respectively, to investigate the iCT process in detail. The FT-IR spectrum of **TMTQ** in toluene exhibits the distinguished C≡N stretching IR band at 2214 $cm^{-1}$ (Fig. 2e). This ground electronic state C≡N stretching IR band showed a negligible shift between toluene, $CH_2Cl_2$, and $CH_3NO_2$, which indicates that the electron density on the dicyanomethylene groups is similarly distributed regardless of solvent polarity. This feature is well matched with the steady-state absorption results and suggests that there is no CT character in the ground state.

For an in-depth study of the iCT process in the excited state, the transient IR spectra of **TMTQ** were measured in toluene, $CH_2Cl_2$, and $CH_3NO_2$, where we focused on the region of

2140–2240 $cm^{-1}$ to monitor the change of the C≡N stretching mode (Fig. 2f). The transient IR spectra in the three solvents showed a common GSB signal around 2214 $cm^{-1}$, which is matched with the ground-state IR band for C≡N stretching mode. In addition, a distinct PIA band was obtained around 2170 $cm^{-1}$ regardless of solvent polarity. This transient IR spectral feature indicates that the C≡N stretching IR band is 45 $cm^{-1}$ red-shifted in the excited state. Since the decay of the GSB signal in the transient IR spectra is in accordance with that of TA signal (Supplementary Fig. 5), the observed 45 $cm^{-1}$ red-shift of the C≡N stretching IR band clearly depicts the iCT process of **TMTQ** in the excited state by the shift of π-electron density from the M10A donor to the DT acceptor.

Here, compared to the transient IR spectra of **TMTQ** in toluene, those in $CH_2Cl_2$ and $CH_3NO_2$ displayed an additional PIA band around 2205 $cm^{-1}$, which is a slightly red-shifted from the ground-state C≡N stretching band (Supplementary Fig. 5). This PIA band appeared within the decay of the excited-state C≡N stretching band at 2170 $cm^{-1}$. These features of the PIA band around 2205 $cm^{-1}$ are indicative of hot-vibrational bands of

the C≡N stretching vibrations in the ground state[36], portraying that the relaxation of excited species in the CT state produces thermally excited vibrational motions in the ground state and subsequent vibrational cooling occurs in $CH_2Cl_2$ and $CH_3NO_2$ solvents (Supplementary Fig. 6). These hot-vibrational spectral features are well matched with the additional lifetime component observed in the TA spectra in $CH_2Cl_2$ (15 ps) and $CH_3NO_2$ (19 ps) conditions, where increasing the lifetime for the vibrational cooling process upon the higher solvent polarity well describes the stabilization effect of the CT state by the solvent polarity.

Surprisingly, the excited-state C≡N stretching IR bands at 2170 cm$^{-1}$ in the transient IR spectra are exactly matched with the C≡N stretching IR band of the electrochemically reduced **TMTQ** dianion (Fig. 2g, Supplementary Fig. 7)[24]. This excellent consistency of the C≡N stretching IR bands between the excited and dianionic forms of **TMTQ** indicates that the electron density on the dicyanomethylene groups in the relevant excited state form is similar to that in the dianion. Therefore, it is clearly depicted by the C≡N stretching IR bands that the iCT process in the excited state leads to the shift of two π-electrons from the central M10A donor to both DT acceptors in **TMTQ**, which results in a core annulene with dicationic character possessing 8π-electrons (Fig. 1).

To investigate the conformational change of **TMTQ** by the excited-state iCT process, we measured IR spectra in the region of 1275–1575 cm$^{-1}$ for C=C stretching modes[37,38]. Because the intensity of IR bands is sensitive to the change of dipole moment within the vibrational motions[15,21,39,40], the dipole moment change triggered by the C=C stretching modes along the π-conjugation pathways allows us to estimate structural planarization (or distortion) based on the decreased (or increased) IR intensity of C=C stretching modes between the FT-IR and transient IR spectra. The FT-IR spectra showed intense bands at 1479 and 1529 cm$^{-1}$ and weak bands at 1301, 1359, and 1421 cm$^{-1}$, which are well matched with the GSB signals in the transient IR spectra. Conversely, the transient IR spectra for the C=C stretching vibrational region showed weak PIA bands (except for the vibrational hot bands) in cohabitation with intense GSB bands in the 1430–1550 cm$^{-1}$ region. On the other hand, in the 1275–1410 cm$^{-1}$ range, distinctively intense PIA and weak GSB bands were measured. This suggests a significant difference between the ground-state and excited-state IR spectra and reflects that the iCT process is accompanied by substantial structural changes.

**Conformational change and excited-state aromatization**. The excited-state iCT process in **TMTQ** raises the possibility for aromatization in the excited state. On the basis of Baird's rule, Hückel aromatic (or antiaromatic) rules of $[4n + 2]\pi$ (or $[4n]\pi$) systems in the ground state are reversed in the excited state, where they become antiaromatic (or aromatic)[3]. In **TMTQ**, the shift of two π-electrons from M10A to both sides of the DT units by the iCT process produces a dicationic M10A, which, in the excited state, can lead to a Baird aromatic core annulene with 8π-electrons.

For qualitative investigation of the excited-state aromatization in **TMTQ**, we focused on the iCT-induced conformational change by means of the IR-activity analysis of the relevant C=C stretching vibrational modes (Supplementary Fig. 8). Because the C=C stretching motions primarily vibrate along the π-conjugation pathway, their IR-activities are sensitive to structural distortion[20]. The C=C stretching motions in aromatic skeletons take place along planar and symmetric molecular frameworks carrying out negligible (null) change of dipole moment, which leads to very weak (forbidden) IR absorption bands. In contrast,

when these C=C stretching motions occur along distorted molecular skeletons, they gain an out-of-plane contribution that triggers a significant dipole moment change and become more IR-allowed.

For an in-depth analysis of C=C stretching IR bands, we extracted the excited-state IR spectra from the transient IR spectra and compared them with the ground-state IR (FT-IR) spectrum (Supplementary Figs. 9 and10). The intense bands in the region of 1450–1550 cm$^{-1}$ of the ground-state IR spectrum were attenuated in the excited-state IR spectra. On the other hand, compared to the weak bands in the region of 1270–1450 cm$^{-1}$ in the ground-state IR spectrum, distinctively intense bands were obtained in this region in the excited-state IR spectra. These sharp contrasting IR spectral features between the ground and excited states well illustrate that the iCT process induces a considerable conformational alteration of **TMTQ** in the excited state.

In the comparative analysis between the experimental and calculated IR spectra (Fig. 3), the IR spectrum calculated for the S$_0$ state of **TMTQ** showed excellent consistency with that obtained experimentally. This reveals that the S$_0$ optimized geometry derived from calculations and the true ground electronic state molecular structure of **TMTQ** are certainly close. To obtain valuable theoretical data for IR spectra in the excited states, TD-DFT calculations were carried out in the S$_1$ and T$_1$ states. These two lowest lying related excited states are well known to be structurally similar in closed-shell molecules. On the other hand, the accurate prediction of spectroscopic properties in the excited states of π-conjugated molecules remains a challenge for quantum chemical methods, a situation which is particularly difficult for the elucidation of excited-singlet states and significantly ameliorated in the corresponding triplet state due to the distinctive electron-electron correlation. Hence, it is sometimes preferred to consider TD-DFT calculations of the excited triplet state to understand the homolog excited-singlet state rather than conduct them on the singlet state itself[21]. Going to the current results in our study, the experimental transient IR spectrum of **TMTQ** is compared with those obtained for the S$_0$ and T$_1$ states (Supplementary Figs. 11–13) from which we observe that, in line with the discussion above, the resemblance to the T$_1$ state is better than that for the S$_1$ state, from which we consider the former to qualitatively understand the changes in the transient IR experimental spectrum and use these results to guide and address the qualitative changes in molecular geometries and conformations in the excited two-electron CT state.

In the calculated S$_0$-state IR spectrum, the sharp and distinct bands at 1512, 1537, and 1563 cm$^{-1}$ with several weak bands in the region of 1250–1475 cm$^{-1}$ lead to an excellent consistency with the ground-state IR spectrum. The vibrational mode analysis revealed that the IR-active C=C stretching vibrational modes at 1512, 1537, and 1563 cm$^{-1}$ accompany the intense stretching along the core annulene (Fig. 3c, e). On the other hand, the IR-inactive C=C stretching modes at 1276, 1281, 1389, and 1453 cm$^{-1}$ showed the weak stretching motions along the long molecular axis between the dicyanomethylene groups. For the calculated T$_1$-state IR spectrum, intensity-attenuated bands at 1497, 1534, and 1560 cm$^{-1}$ together with intensified bands at 1296, 1307, 1387, and 1433 cm$^{-1}$ were obtained, which is well matched with the excited-state IR spectrum (Fig. 3d, f). Whereas the vibrational modes at 1497, 1534, and 1560 cm$^{-1}$ exhibited the C=C stretching motions concentrated on the core annulene, the C=C stretching vibrations along the long-axis of **TMTQ** are represented by the IR-active vibrational modes at 1296, 1307, 1387, and 1433 cm$^{-1}$.

This vibrational mode analysis elucidates the conformational change of **TMTQ** by the iCT process. In the ground state, **TMTQ** possesses the distorted M10A moiety while the DT moieties are linearly aligned at both sides (Fig. 4). In this geometry, the C=C stretching vibrations within the twisted core annulene contain an

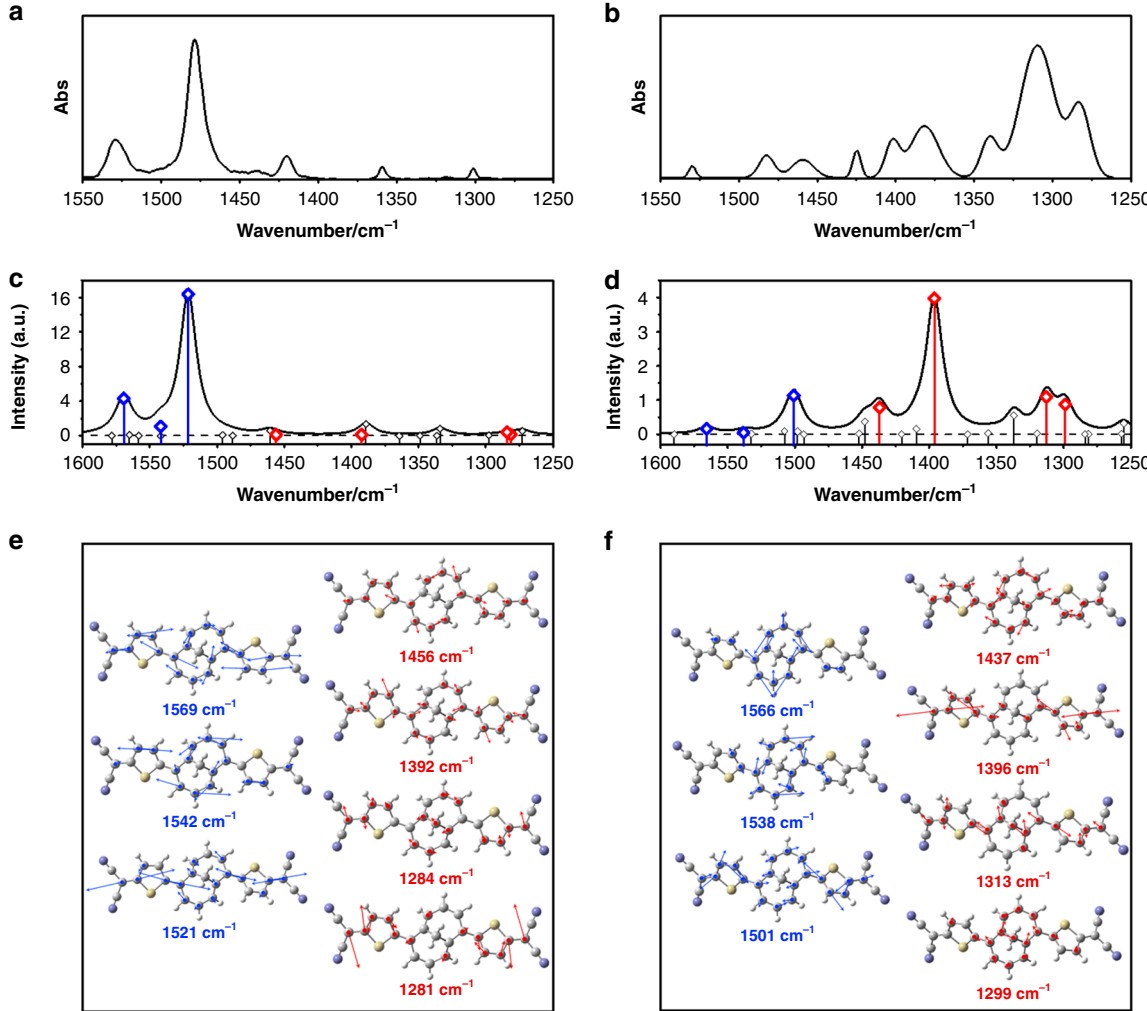

**Fig. 3** Analyses of C=C stretching IR spectral features of TMTQ. **a–d** The experimental ground-state (**a**) and excited-state (**b**) IR spectra and calculated $S_0$-state (**c**), and $T_1$-state (**d**) IR spectra of TMTQ. In the calculated IR spectra, blue-colored vertical bars indicate the C=C stretching modes along the core annulene and red colored bars correspond to the C=C stretching vibrations over the long axis of TMTQ. **e–f** The C=C stretching vibrational motions of TMTQ in the $S_0$ (**e**) and $T_1$ states (**f**) are colored as blue and red, respectively, which are matched with the corresponding C=C stretching vibrational modes in the calculated IR spectra

out-of-plane component that produces a significant dipole moment change along the vibrational motions, resulting in IR-active 1450–1550 cm$^{-1}$ vibrations. On the other hand, the stretching motions over the long-axis of **TMTQ** with small motions on the core annulene produce weak IR bands in the region of 1250–1450 cm$^{-1}$, where the relatively linear geometry of **TMTQ** provokes a minor change of dipole moment that renders these vibrations IR inactive. The optimized structure of **TMTQ** in the $T_1$ state showed the [10]annulene moiety with significantly released distortion and relatively planar geometry despite the concomitant strain on the methylene bridge in M10A. Here, the out-of-plane components of the stretching vibrations along the core annulene are attenuated and the absorbances of the associated IR bands thus become reduced. Conversely, in the whole bent geometry, the C=C stretching vibrations along the long axis of **TMTQ** acquire out-of-plane components, resulting in the enhanced IR-intensity features. For more deliberate interpretation for the IR spectral changes, we have also analyzed the C=C stretching IR bands of triplet M10A dication fragment in various degrees of structural distortion (Supplementary Fig. 14), where the quinoidal nature of **TMTQ** system with its aromaticity change was well reflected by the triplet dicationic character. The

TD-DFT frequency results showed that, upon the structural planarization with enhanced aromatic nature, the C=C stretching IR intensity of M10A dication becomes significantly attenuated, which is well matched with our experimental observations. Therefore, the observed stark contrast in the IR spectral features in the region between 1250–1450 and 1450–1550 cm$^{-1}$ is originated from the iCT-induced conformational change of **TMTQ** from the distorted core annulene in the whole linear structure to the planar one in the whole curved structure, which well delineates the aromaticity-driven planarization of core 8π annulene in the CT state.

For more comprehensive understanding about the iCT-induced conformational change, we quantitatively analyzed the TD-DFT optimized structures of **TMTQ**. In the ground ($S_0$) state, the optimized structure of **TMTQ** exhibited a severe distortion in the central M10A moiety (Fig. 4a), producing a pronounced bond length alternation pattern between the C-C and C=C bonds (Fig. 4b,, Supplementary Fig. 15) in line with the non-aromatic nature of 10π core annulene. Anisotropy of the induced current density (ACID) analysis well portrayed this discrete π-conjugation of **TMTQ** (Supplementary Fig. 16)[41,42] and suggests that the appended DT units impose discrete $S_0$ state π-

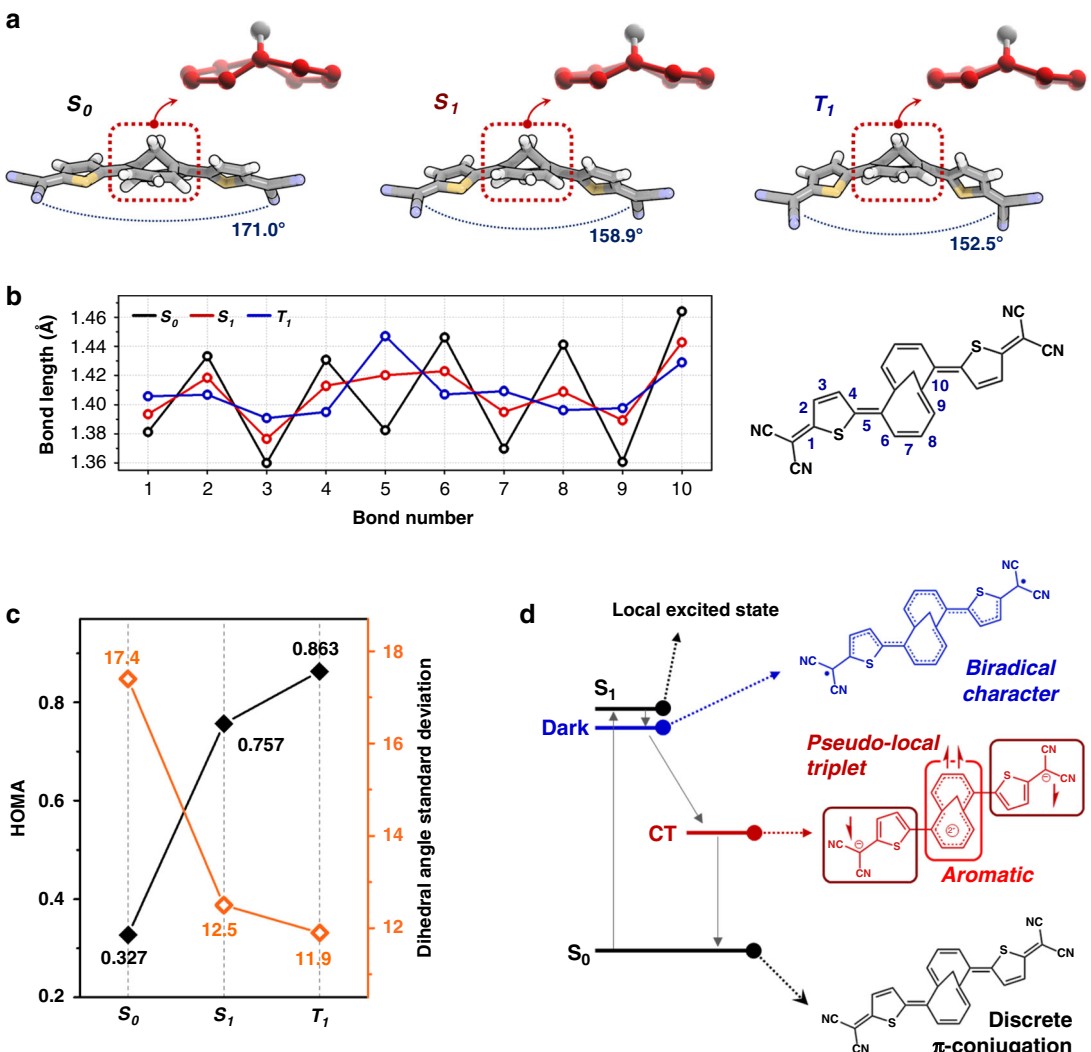

**Fig. 4** Excited-state dynamics and conformation analyses. **a** The (TD)DFT-based optimized structure of TMTQ in the $S_0$, $S_1$, and $T_1$ states. The core annulene is colored as red. **b** The carbon–carbon bond length plot for TMTQ. The molecular structure is drawn based on the bond length distribution in the $S_0$ state. **c** The plot for HOMA and dihedral angle standard deviation values of core annulene. **d** Schematic illustration for the electronic structures and excited-state aromatization

conjugation. The bond length alternation is reduced in the optimized structures for the $S_1$ and $T_1$ states. In particular, in the $T_1$ state of **TMTQ** the M10A moiety becomes more planar and symmetric. In the bond length plot, the bond length deviation certainly decreased except the significant elongation of the annulene-thiophene bond. This structural change describes that the π-conjugation is extended compared to the $S_0$ state and the local conjugative interaction on each M10A and DT moiety is enhanced.

To quantitatively evaluate the aromatic character of the core annulene upon the conformational change of **TMTQ**, we estimated the harmonic oscillator model of aromaticity (HOMA) and the dihedral angle standard deviation (DASD; Fig. 4c)[43]. The HOMA value quantitatively describes the π-electron delocalization in cyclic π-conjugated molecules based on bond lengths, where the value of 1 (or 0) indicates completely aromatic (or non-aromatic) molecules. The HOMA values for the core annulene increased going from the $S_0$ to $S_1$ and $T_1$ state (0.281 → 0.725 → 0.841). As for the DASD values of the annulene core, these values reduced from the $S_0$ to $S_1$ and $T_1$ state (20.8 → 12.5 → 11.9). Since the dihedral angle represents a local structural distortion, the decrease in DASD value describes the structural planarization

of core annulene. Therefore, the dramatic increase of HOMA values together with the decrease of DASD values between the optimized structure for the $S_0$ and $T_1$ states well portray the change of the non-aromatic character of the core 10π annulene in the $S_0$ state into the effectively conjugated excited state 8π annulene possessing strong aromatic character in the CT state.

## Discussion

As mentioned, Baird's rule describes the reversed aromaticity in the excited triplet state, which reminds that the core 8π annulene of **TMTQ** should be triplet in the CT state. However, our experimental observation showed an absence of intersystem crossing in the excited-state dynamics of **TMTQ**. This suggests that other spin configurations are involved in the excited-state aromatization, where π-electrons on the M10A and DT moieties play an important role in the overall excited-state electronic structures.

For an exhaustive analysis of excited-state dynamics of **TMTQ**, we explored the electronic structures by multi-configurational electronic structure (RAS-SF-srDFT) calculations[44,45]. These quantum chemical analyses revealed that in the Franck–Condon region there are two main low-energy lying singlet excited states

($S_1$ and dark states in Fig. 4d) which play a significant role in the excited-state dynamics of **TMTQ**. The $S_1$ and dark states are optically allowed and optically forbidden, respectively, and between them there is a substantially small energy gap (0.08 eV). The comparative analysis of TD-DFT and RAS-SF-srDFT calculation results for the lowest excited-singlet state, we discovered an excited-singlet state with a rather strong solvent dependence, pointing toward a presence of the CT state, which are energetically more stable than the dark state. These calculation results for electronic states of **TMTQ** are well matched with the experimental observations (Supplementary Fig. 6). The calculated optically allowed $S_1$ and optically forbidden dark states with the small energy gap of 0.08 eV well address the ultrafast initial decay (>40 fs) in the TA decay profile. Moreover, the presence of the iCT state rationalizes the solvent-polarity dependence observed in the excited-state dynamics of **TMTQ** with the 45 cm$^{-1}$ red-shifted C≡N stretching IR band in the transient IR spectra.

In the electronic structural analysis, it is discovered that this dark state is mainly composed of double HOMO-LUMO excitations, thus manifesting its optically forbidden and multiexcitonic nature (Supplementary Fig. 17). The multiexcitonic character can be understood by biradical effect (Fig. 4d)[46–50]. In contrast to the discrete π-conjugation of **TMTQ** in the $S_0$ state, **TMTQ** in the dark state can possess an extended π-conjugation from the thiophene to the core annulene moieties, which can justify the reduced bond length alternation found from the ground to the excited state (Fig. 4b, Supplementary Fig. 18). This enhanced conjugative interaction stabilizes the formation of radicals on each dicyanomethene group, thus resulting in two strongly coupled local excitations separated on each terminal of **TMTQ**[15,51], which is a well-known multiexcitonic feature in carotenoid molecules[52]. As sketched in Fig. 4d, the subsequent transition from the dark to CT state facilitates a pseudo-local triplet configuration in the embedded singlet multiexcitonic nature. In this local triplet-like configuration, the iCT-induced 8π annulene becomes aromatic based on Baird's rule. We found that a planarized dicationic M10A shows distinct aromatic nature in the triplet state by the ACID and NICS analyses (Supplementary Fig. 19). Therefore, the large energetic advantage from the triplet Baird aromatic stabilization mediates the singlet multiexcitonic nature to be coupled to the CT state, which effectively stabilizes the two equivalents of charge separated **TMTQ** with the pseudo-local triplet configuration on the core annulene.

Here, it is suggested that the excited-state aromatization mediated by the biradical character in **TMTQ** is a driving force for the shift of two π-electrons with the multiexcitonic CT state. Typically, the iCT process accompanying two π-electron transfer is unusual due to a large energetic disadvantage. However, in **TMTQ**, the excited-state aromatization in the multiexcitonic state together with the biradical effect and the electron-withdrawing nature of the dicyano groups exceptionally stabilizes the unfavorable transfer of two π-electrons. This advocates that the excited-state aromaticity can play a key role in dictating electronic nature in the excited states, which is of tremendous value for the exploitation of D–A or A–D–A systems.

In summary, we consistently described the intramolecular CT process driven by excited-state aromatization in **TMTQ**. The optical spectroscopic observation clearly verified the shift of two π-electrons in M10A moiety to each DT moiety in the iCT process, driven by the aromatization of 8π core annulene in the excited state. Quantum chemical analyses revealed the multiexcitonic character arising from the biradical character on the terminal groups allowed the core annulene to accommodate the local triplet-like configuration, thus leading to the excited-state aromatization and stabilization of the two-charge separated state. This finding leads to in-depth understanding of the role of excited-state aromaticity in the excited-state dynamics and the underlying mechanism will provide a fruitful insight into a rational design of complex excitonic properties in organic electronic materials.

## Methods

**Steady-state and time-resolved optical characterization.** The steady-state electronic and IR absorption spectra of **TMTQ** were measured with a UV-Vis-NIR spectrometer (Varian, Cary 5000) and FT-IR spectrometer (Bruker, Vertex 70) in toluene, $CH_2Cl_2$, and $CH_3NO_2$. The IR spectro-electrochemical data were obtained by using an optically transparent thin layer electrochemical (OTTLE) cell. The working electrode (Pt minigrid, 32 wires/cm) potential was controlled with an Electrochemical Analyzer BAS 100B and referenced to the Fc/Fc$^+$ couple. The supporting electrolyte was a 0.1 mol/L dichloromethane solution of tetra-butylammonium hexafluorophosphate (TBA-PF$_6$). In OTTLE cell, optical path and sample volume is less than 0.2 mm and 0.2 mL, respectively. The transient electronic and IR absorption data were obtained by the home-made pump-probe method spectrometers constructed with femtosecond regeneratively amplified Ti: sapphire laser systems and UV-Vis-NIR/mid-IR detectors, where all the measurements were conducted under the solvent-flowing condition to prevent photo-degradation. The full information for the optical characterization is described in Supplementary Methods.

**Quantum chemical calculations.** Quantum mechanical calculations of **TMTQ** for molecular geometry optimization and vibrational frequency analysis were performed by the Gaussian09 Revision E.01 program suite[53] (for full reference, see Supplementary Reference). Quantum mechanical calculations for in-depth analyses of electronic structures have been done with the Q-Chem program[54]. More detailed information for the whole computational analyses was described in Supplementary Methods and Tables 2–5.

## Data availability

All other data supporting the findings of this study are available within the article and its Supplementary Information.

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

## Acknowledgements

This research at Yonsei University supported by the Strategic Research (NRF2016R1E1A1A01943379) through the National Research Foundation of Korea (NRF) funded by the Ministry of Science, and related quantum calculations were performed with the supercomputing resources of the Korea Institute of Science and Technology Information. We thank MINECO/FEDER of the Spanish Government (CTQ2015-69391-P and CTQ2016-80955-P) projects and the US National Science Foundation (CHE-1607821). The work at Pusan National University was supported by the National Research Foundation of Korea (NRF) grant funded by the MEST (NRF-2017R1A2A2A05001052).

## Author contributions

J.O. and D.K. conceived and conceptualized the work. J.O. and J.K. performed the time-resolved spectroscopy and analyzed the data together with experimental design and supervision provided by D.K. S.P. and J.L.Z. carried out vibrational experiments. J.R.D. synthesized the target system. D.C. carried out the quantum chemical calculations. J.O., M.L., J.C., J.D.T., D.C., and D.K. discussed the scientific contents and co-wrote the paper.

## Competing interests

The authors declare no competing interests.
