## [Peer Review File · Nature Communications]

Reviewers' comments:

Reviewer #1 (Remarks to the Author):

Having read carefully the authors' responses to the reviewers' comments on the previous manuscript and the new version of the manuscript, I find that the authors have managed to address all major concerns of the reviewers and have invested a significant amount of time and effort in providing a new set of computational results which show improved agreement with experimental data. In my opinion, the novelty elements in this manuscript, both in terms of experimental and computational results, are sufficient to justify publication in Nature Communications despite the existence of previous studies of TMTQ, *Angew. Chem. Int. Ed.* 2015, 54, 5888 (Ref 24 in the new version) and *Chem. Eur. J.* 2016, 22, 2793 (Ref 51 in the new version). Clearly, as a compound which could become important in pi-conjugated spin-bearing materials, TMTQ is starting to attract considerable interest, and the current manuscript makes a major contribution to the understanding of its excited state behaviour.

Reviewer #2 (Remarks to the Author):

I reviewed the original Nat. Chem. submission of this work (as Referee 2). This time, I started by reading comments of the other referees and found their criticism consistent with mine, both in terms of the originality claims and technical correctness. The authors did a very thorough job revising the manuscript and preparing the rebuttal.

The majority of my concerns were addressed, however, I remain skeptical about the accuracy of the IR analysis for the triplet state. Ignoring the most intense band is a risky move, and the authors provide no convincing rationale for such a selective omission. Furthermore, as the triplet geometry comes from a time-independent UDFT calculation, I wonder why the authors insist that their presumed triplet obtained by photoexcitation is different from the thermally populated triplet discussed in the original ref. 20. I also feel that insufficient emphasis is laid on the triplet controversy (new ref. 51), which is obviously related to the present discussion.

My view on the originality of this work remains unchanged. The revised manuscript is surely closer to meet the acceptance criteria of Nat. Comm. in terms of its potential significance and broad interest. However, the spectroscopic analysis of the triplet state remains inconclusive.

Reviewer #4 (Remarks to the Author):

I have carefully read the revised manuscript, SI, and the authors' careful responses to all of the referees.

Whilst I think that the data themselves are very nice, I still have concerns about their interpretation.

In the authors' response to referee 1's question about Mulliken population analyses, it becomes clear that the computational data and experimental results do not present a fully consistent picture. Since the Mulliken analysis doesn't show charge-transfer in T1, it is of no surprise that the calculated IR spectrum of the T1 state is also in poor agreement with experiment – it just seems like there is excessive delocalization in the DFT picture. The TRIR is unequivocal, based on the CN str, that there is some CT character. But, once we no longer trust the calculated T1 state's charge distribution, I fear that we cannot trust much else: we can't trust the bond length alternation plots in Fig 4 (or the HOMA/dihedrals), nor the calculated IR. Since much of the ms focuses on demonstrating the aromaticity in the CT T1 state based on computational metrics, this seems to

me an unfortunate flaw.

I would suggest that the poor agreement of Mulliken populations may arise from the choice of B3LYP as a functional, whereas a range-separated functional may be more appropriate for this CT case.

All of the referees were concerned by the poor match of the calculated and experimental T1 IR spectra. I think that the authors' explanation that the band at ca. 1400 cm^{-1} is an artefact is helpful. However, even excluding that band, the match is not great and does not feel good enough to provide such a key pillar of evidence for the claims of the ms. Perhaps another functional more suited to CT states may help? Alternatively, it would perhaps be possible (though certainly more complex and challenging) to divide the molecule into fragments and treat those individually (e.g. the M10A core dication, or M10A-thiophene dication). In such a circumstance, comparisons of the single-electron transfer case (ie M10A[-thiophene] cation doublet) to the dication could be helpful. Realistically this fragmentation approach may be too challenging, since the choice of how/where to break up the molecule is quite uncertain.

In their response to Referee 3's second query, the authors have provided a Figure showing their spectroelectrochemical setup. The problems of diffusion/spectral purity can be avoided by aiming the FTIR laser through the working electrode grid. The authors seem (based on Figure 3-1) to have aimed their beam through a region of electrolyte, rather than through the electrode. There should be no particular deleterious effect on the FTIR spectra from the presence of the grid, but much purer spectra (ie pure oxidation states) will be obtained. It should certainly be possible to measure pure dianion by holding a potential beyond that at which the dianion is generated – measurement of the monoanion may require multicomponent curve resolution, or may be practically impossible if the monoanion spectrum is a linear combination of those for the neutral and dianion states.

I remain to be entirely convinced that two electrons transferred in the photophysical experiments, and not just one. Based on the authors' explanation, the ground-state spectrum of 'dianion' in Fig 2g (left, lower) is actually probably monoanion, or at best a mixture of neutral, dianion, and monoanion. It is certainly **not** pure dianion! Any dianion formed at the working electrode would disproportionate by the time it got to the region measured by the FTIR (shown in Fig 3-1). It would be good to repeat the spectroelectrochemical measurements to obtain pure spectra of monoanion and dianion, and use these to assign the TRIR spectrum in Fig 2g (top, left). The most compelling assignment would come from the use of both peak positions and relative intensities/oscillator strengths.

Response to Reviewers' Comments

First of all, we are sincerely appreciative of the reviewers who have obviously taken considerable time out of their own busy lives to help us make this a better paper through their thoughtful consideration and advice. Our responses and specific revisions to the reviewers' comments are shown below.

[Response to Reviewer 1]

The comment from Reviewer 1: *Having read carefully the authors' responses to the reviewers' comments on the previous manuscript and the new version of the manuscript, I find that the authors have managed to address all major concerns of the reviewers and have invested a significant amount of time and effort in providing a new set of computational results which show improved agreement with experimental data. In my opinion, the novelty elements in this manuscript, both in terms of experimental and computational results, are sufficient to justify publication in Nature Communications despite the existence of previous studies of TMTQ, Angew. Chem. Int. Ed. 2015, 54, 5888 (Ref 24 in the new version) and Chem. Eur. J. 2016, 22, 2793 (Ref 51 in the new version). Clearly, as a compound which could become important in pi-conjugated spin-bearing materials, TMTQ is starting to attract considerable interest, and the current manuscript makes a major contribution to the understanding of its excited state behaviour.*

Our response: We sincerely appreciate the reviewer's comprehensive understanding of our scientific findings with his/her great interest. As mentioned in the reviewer's comments, we have also concentrated on revealing an effect of excited-state aromaticity and strongly believe that our study provides a new and crucial insight into the understanding of excited-state behaviors and their applications. Therefore, we are grateful to the reviewer's positive comments and full attention on our study.

[Response to Reviewer 2]

The comment from Reviewer 2: *I reviewed the original Nat. Chem. submission of this work (as Referee 2). This time, I started by reading comments of the other referees and found their criticism consistent with mine, both in terms of the originality claims and technical correctness. The authors did a very thorough job revising the manuscript and preparing the rebuttal.*

The majority of my concerns were addressed, however, I remain skeptical about the accuracy of the IR analysis for the triplet state. Ignoring the most intense band is a risky move, and the authors provide no convincing rationale for such a selective omission. Furthermore, as the triplet geometry comes from a time-independent UDFT calculation, I wonder why the authors insist that their presumed triplet obtained by photoexcitation is different from the thermally populated triplet discussed in the original ref. 20. I also feel that insufficient emphasis is laid on the triplet controversy (new ref. 51), which is obviously related to the present discussion.

My view on the originality of this work remains unchanged. The revised manuscript is surely closer to meet the acceptance criteria of Nat. Comm. in terms of its potential significance and broad interest. However, the spectroscopic analysis of the triplet state remains inconclusive.

Our response: We are glad that the reviewer recognized our efforts to deliver clearer description of our scientific findings. And we also sincerely thank the reviewer for letting us know what we have missed and need to explain more. First, for the reviewer's concern on our IR analysis based on TD-DFT calculations, we did not just suggest the exclusion of the most intense IR band around 1400 cm^{-1} simply for a spectral similarity. The IR band around 1400 cm^{-1} was also overestimated in the TD-DFT frequency results for the S_1 state even though TMTQ showed less structural changes in the S_1 state and the other IR bands in the lower energy region ($1250\sim 1400\text{ cm}^{-1}$) exhibited weak IR intensities. To explain this feature, we put Figure S11 in the previous version of supplementary information.

For the reviewer's another concern on the triplet state, we appreciate the reviewer's calling this issue to our attention and are sorry to have confused you with our rather insufficient explanation. In this study, we experimentally observed the two electron transfer process, as all reviewers conceded. Thus, what we experimentally observed and focused on is not the triplet state but the excited CT state. In our experimental results, all the excited state dynamics and processes occurred rapidly with the time constants of less than tens of picoseconds, which is too fast to consider the intersystem crossing between singlet and triplet states as long as there is no heavy-metal effect. We also described this point in our

Figure S11. The calculated IR spectra of **TMTQ** for the S_0 (a), S_1 (b) and T_1 (c) states and C=C stretching vibrational motions in the S_0 (d), S_1 (e) and T_1 (f) states (All data obtained by B3LYP-D3/6-311G(d,p)). Compared to the S_0 -state and experimental IR spectra, the red-marked S_1 - and T_1 -state IR bands around 1400 cm^{-1} , arising from their enhanced conjugation along the long-axis in the linear geometry of **TMTQ** in the S_1 and T_1 states, are overestimated.

previous version of manuscript, “As mentioned, Baird’s rule describes the reversed aromaticity in the excited triplet state, which reminds that the core 8π annulene of **TMTQ** should be triplet in the CT state. However, our experimental observation showed an absence of intersystem crossing in the excited-state dynamics of **TMTQ**. This suggests that other spin configurations are involved in the excited-state aromatization, where π -electrons on the M10A and DT moieties play an important role in the overall excited-state electronic structures.” (15 page, line 8 in the manuscript)

Furthermore, the excited CT state, that we proposed in which Baird aromaticity takes place, is a singlet excited state which resides at around +1.8 eV over the singlet ground electronic state. However, the true T_1 state detected by heating from the ground state in reference 24 is at around +0.15 eV over the ground electronic state. However, the joint discussion on these different excited singlet and triplet states that might have confused the reviewer is argued in the following way: 1) after electronic structure calculations at the RAS-2SF-srPBE level of theory we conclude that the singlet CT state can be described by the coupling of two correlated triplets in which one of them is locally placed on the central annulene core (but this state is overall a singlet). It is on this local pseudo triplet that Baird aromaticity applies. This local pseudo triplet state on the core annulene might resemble the structure of the heating accessible true T_1 state BUT the involved states are largely different!. 2) Based on the previous

similitude on (1) and also based on the argument on line 9, page 11 in the revised version of the manuscript (see below) we attempt to use calculations on the true T_1 state to QUALITATIVELY understand the vibrational properties of the excited CT state only.

In fact, the electronic structure of **TMTQ** in its singlet excited state manifold (not triplet manifold) was computationally analyzed with the multi-configurational calculation with RAS-2SF-srPBE functional as well as the TD-DFT calculation (see section 2.4 in Figure 4d in the revised manuscript). Although multi-configurational calculations show higher accuracy in the electronic structure, they are not optimized to vibrational frequency analysis. Hence, TD-DFT frequency results of **TMTQ**, that show an excellent consistency with the experimental data in the ground state at the B3LYP-D3/6-311G(d,p) level, were obtained for the S_1 and T_1 states and compared with the experimental excited-state IR spectra. It is here that we use the TD-DFT vibrational infrared calculations on the T_1 state to QUALITATIVELY interpret the experimental excited state IR spectrum. This approach has been already used for the excited singlet state (S_n) in our recent study (*Chem* **2017**, *3*, 870-880) empirically suggested that, for the excited singlet state, the larger conformational changes, being more close to the T_1 -state optimized structures rather than S_1 -state ones, are expected.

Taking all these points into consideration, our IR analysis with the TD-DFT frequency results for the T_1 state is considered to be reliable for interpreting our experimental IR data and revealing the conformational changes of **TMTQ** in the excited states. To deliver this information more perspicuously, we changed the related parts in the revised manuscript. Once again, we sincerely thank the reviewer for helping us communicate more clearly in our manuscript.

Regarding the controversy described in reference 51, we are aware that the triplet discussed in this article is the true T_1 state (thermally accessible from the ground electronic state) which is characterized by a 15% Baird aromaticity contribution according to the authors (they defined it as a Hückel-Baird hybrid). In the same article, the authors described that for higher excited states the contribution from Baird aromaticity will progressively increase and this is actually our case in which the maximal Baird aromaticity contribution is in fact prepared by previous two electron transfer and generation of the CT state.

>>>(Page 11, line 7 in the revised manuscript)

In the comparative analysis between the experimental and calculated IR spectra (Fig.3), the IR spectrum calculated for the S_0 state of **TMTQ** showed excellent consistency with that obtained experimentally.

This reveals that the S_0 optimized geometry derived from calculations and the true ground electronic state molecular structure of **TMTQ** are certainly close. To obtain valuable theoretical data for IR spectra in the excited states, TD-DFT calculations were carried out in the S_1 and T_1 states. These two lowest lying related excited states are well known to be structurally similar in closed-shell molecules. On the other hand, the accurate prediction of spectroscopic properties in the excited states of π -conjugated molecules remains a challenge for quantum chemical methods, a situation which is particularly difficult for the elucidation of excited singlet states and significantly ameliorated in the corresponding triplet state due to the distinctive electron-electron correlation. Hence, it is sometimes preferred to consider TD-DFT calculations of the excited triplet state to understand the homologue excited singlet state rather than conduct them on the singlet state itself. Going to the current results in our study, the experimental transient IR spectrum of **TMTQ** is compared with those obtained for the S_0 and T_1 states (Fig. S10-12) from which we observe that, in line with the discussion above, the resemblance to the T_1 state is better than that for the S_1 state, from which we consider the former to qualitatively understand the changes in the transient IR experimental spectrum and use these results to guide and address the qualitative changes in molecular geometries and conformations in the excited two-electron CT state.

[Response to Reviewer 4]

First comment from Reviewer 4: *I have carefully read the revised manuscript, SI, and the authors' careful responses to all of the referees.*

Whilst I think that the data themselves are very nice, I still have concerns about their interpretation.

In the authors' response to referee 1's question about Mulliken population analyses, it becomes clear that the computational data and experimental results do not present a fully consistent picture. Since the Mulliken analysis doesn't show charge-transfer in T_1 , it is of no surprise that the calculated IR spectrum of the T_1 state is also in poor agreement with experiment – it just seems like there is excessive delocalization in the DFT picture. The TRIR is unequivocal, based on the CN str, that there is some CT character. But, once we no longer trust the calculated T_1 state's charge distribution, I fear that we cannot trust much else: we can't trust the bond length alternation plots in Fig 4 (or the HOMA/dihedrals), nor the calculated IR. Since much of the ms focuses on demonstrating the aromaticity in the CTT_1 state based on computational metrics, this seems to me an unfortunate flaw.

Our response: We appreciate the reviewer's attention on our study and recognition of our experimental results. We have already discussed this point with the reviewer 2 and are sorry for our short explanation to confuse the reviewers. In our study, what we have discussed and experimentally detected is not the triplet state but the excited CT state. Our experimental measurements showed the rapid excited state dynamics with the time constant of less than tens of picoseconds, which is too fast to consider intersystem crossing between singlet and triplet states as long as there is no heavy-metal effect. This point is also mentioned in our previous version of manuscript, "*As mentioned, Baird's rule describes the reversed aromaticity in the excited triplet state, which reminds that the core 8π annulene of **TMTQ** should be triplet in the CT state. However, our experimental observation showed an absence of intersystem crossing in the excited-state dynamics of **TMTQ**. This suggests that other spin configurations are involved in the excited-state aromatization, where π -electrons on the M10A and DT moieties play an important role in the overall excited-state electronic structures.*" (15 page, line 8 in the manuscript) To verify this suggestion, we computationally analyzed the electronic structures of **TMTQ** with the multi-configurational calculations with RAS-2SF-srPBE functionals as well as the TD-DFT calculations. As the reviewer mentioned, because the TD-DFT calculation results alone did not support the observed electronic structures of **TMTQ**, the comparative analysis with the multi-configurational calculations provides highly accurate results, which was comparatively analyzed with our experimental

data and finally verified the electronic structures of **TMTQ** in the section 2.4 of manuscript with Figure 4d.

Here, we insist on the fact that we relied on the calculations on the T_1 state ONLY for an interpretation of the vibrational spectra. Although multi-configurational calculations show a higher accuracy in the electronic structure analysis, they are poor in analyzing conformational changes and vibrational frequencies. On the other hand, the TD-DFT calculations provide highly accurate results for the ground state conformations and vibrational frequencies. In our study, the TD-DFT frequency results of **TMTQ** in the ground state from B3LYP-D3/6-311G(d,p) show an excellent consistency with the experimental data. In this regard, we compared the experimental excited-state IR spectra with the TD-DFT frequency results for the S_1 and T_1 states for an in-depth investigation of the experimentally observed IR spectral changes. In particular, the reproduced similar IR spectral features, the contrasting IR intensities between lower and higher energy region, in the TD-DFT frequency results for the T_1 state assisted our qualitative interpretation of experimental IR data. In the same line, for more detailed information for the IR spectral changes, we also comparatively analyzed the TD-DFT optimized structures in the S_0 , S_1 and T_1 states with HOMA and dihedral angle analysis methods. All remaining experimental and computational data were described in the manuscript for the excited CT state. It must be highlighted that the excited CT and T_1 states are different.

In addition, the excited singlet state (S_n) calculations are still a challenging issue in the quantum chemical calculations. Thus, the TD-DFT optimized structures for the S_n states are incomplete. Moreover, the recent study (*Chem* **2017**, 3, 870-880) empirically suggested that, for the excited singlet state, the larger conformational changes, being more close to the T_1 -state optimized structures rather than S_1 -state ones, are expected. Taking these points into consideration, our IR analysis with the TD-DFT frequency results for the T_1 state is considered to be reliable for QUALITATIVELY interpreting our experimental IR data and revealing the conformational changes of **TMTQ** in the excited state. To deliver this information more precisely, we modified the related parts in the revised manuscript. Once again, we sincerely thank the reviewer for helping us communicate more clearly in our manuscript.

>>>(Page 11, line 7 in the revised manuscript)

In the comparative analysis between the experimental and calculated IR spectra (Fig.3), the IR spectrum calculated for the S_0 state of **TMTQ** showed excellent consistency with that obtained experimentally. This reveals that the S_0 optimized geometry derived from calculations and the true ground electronic

state molecular structure of **TMTQ** are certainly close. To obtain valuable theoretical data for IR spectra in the excited states, TD-DFT calculations were carried out in the S_1 and T_1 states. These two lowest lying related excited states are well known to be structurally similar in closed-shell molecules. On the other hand, the accurate prediction of spectroscopic properties in the excited states of π -conjugated molecules remains a challenge for quantum chemical methods, a situation which is particularly difficult for the elucidation of excited singlet states and significantly ameliorated in the corresponding triplet state due to the distinctive electron-electron correlation. Hence, it is sometimes preferred to consider TD-DFT calculations of the excited triplet state to understand the homologue excited singlet state rather than conduct them on the singlet state itself. Going to the current results in our study, the experimental transient IR spectrum of **TMTQ** is compared with those obtained for the S_0 and T_1 states (Fig. S10-12) from which we observe that, in line with the discussion above, the resemblance to the T_1 state is better than that for the S_1 state, from which we consider the former to qualitatively understand the changes in the transient IR experimental spectrum and use these results to guide and address the qualitative changes in molecular geometries and conformations in the excited two-electron CT state.

>>>(Page 14, line 11 in the revised manuscript)

For more comprehensive understanding about the iCT-induced conformational change, we quantitatively analysed the TD-DFT optimized structures of **TMTQ**.

Second comment from Reviewer 4: *I would suggest that the poor agreement of Mulliken populations may arise from the choice of B3LYP as a functional, whereas a range-separated functional may be more appropriate for this CT case.*

Our response: We deeply appreciate the reviewer's full attention and supportive advice on our study. According to the reviewer's careful guidance, we checked the TD-DFT calculation results from various range-separated functionals, CAM-B3LYP and a series of M06 functionals (M06, M06-L, M06-2X and M06-HF). Mulliken population and $C\equiv N$ stretching IR frequency results showed a similar change in charge distribution and no significant difference was observed in the calculation results with the range-separated functionals (Figure R1-R3). On the other hand, the IR spectral data in the $C=C$ stretching region with the range-separated functionals showed an inconsistency with the experimental results (Figure R4). Based on these results, it is considered that the calculation results with B3LYP-D3/6-

311G(d,p) is most suitable for IR data analysis under the current circumstance.

Figure R1. The experimental and simulated IR spectra for C≡N stretching mode of TMTQ.

B3LYP-D3/ 6-311G(d,p)		A	B	C	CAM-B3LYP-D3/ 6-311G(d,p)		A	B	C	M06-2X/ 6-311G(d,p)		A	B	C
Vacuum	S ₀ state	-0.168	0.336	-0.168	Vacuum	S ₀ state	-0.171	0.342	-0.171	Vacuum	S ₀ state	-0.162	0.324	-0.162
	S ₁ state	-0.190	0.380	-0.190		S ₁ state	-0.189	0.378	-0.189		S ₁ state	-0.196	0.392	-0.196
	T ₁ state	-0.169	0.338	-0.169		T ₁ state	-0.163	0.326	-0.163		T ₁ state	-0.174	0.348	-0.174
	Dianion	-0.883	-0.234	-0.883		Dianion	-0.981	-0.038	-0.981		Dianion	-0.985	-0.030	-0.985
CH ₃ NO ₂	S ₀ state	-0.253	0.506	-0.253	CH ₃ NO ₂	S ₀ state	-0.227	0.454	-0.227	CH ₃ NO ₂	S ₀ state	-0.231	0.462	-0.231
	S ₁ state	-0.281	0.562	-0.281		S ₁ state	-0.240	0.480	-0.240		S ₁ state	-0.242	0.484	-0.242
	T ₁ state	-0.221	0.442	-0.221		T ₁ state	-0.218	0.436	-0.218		T ₁ state	-0.229	0.458	-0.229
	Dianion	-1.002	0.004	-1.002		Dianion	-0.910	-0.180	-0.910		Dianion	-0.915	-0.170	-0.915

Figure R2. Mulliken population analysis of TMTQ.

Figure R3. The calculated C≡N stretching IR spectra of TMTQ in the S₀, S₁, T₁ and dianion states.

Figure R4. The experimental ground state (a) and calculated (b-c) IR spectra of TMTQ in the range of 1250~1700 cm^{-1} .

Third comment from Reviewer 4: *All of the referees were concerned by the poor match of the calculated and experimental T_1 IR spectra. I think that the authors' explanation that the band at ca. 1400 cm^{-1} is an artefact is helpful. However, even excluding that band, the match is not great and does not feel good enough to provide such a key pillar of evidence for the claims of the ms. Perhaps another functional more suited to CT states may help? Alternatively, it would perhaps be possible (though certainly more complex and challenging) to divide the molecule into fragments and treat those individually (e.g. the M10A core dication, or M10A-thiophene dication). In such a circumstance, comparisons of the single-electron transfer case (ie M10A[-thiophene] cation doublet) to the dication could be helpful. Realistically this fragmentation approach may be too challenging, since the choice of how/where to break up the molecule is quite uncertain.*

Our response: We are grateful to the reviewer's concern with a sincere advice on our work, "Two-

electron transfer stabilized by excited-state aromatization”. According to the reviewer’s kind guidance, we deliberated the computational analysis of individual fragments. As the reviewer mentioned, “Realistically this fragmentation approach may be too challenging, since the choice of how/where to break up the molecule is quite uncertain”, this approach required a careful choice of fragments because a wrong choice of fragments can give false information. In particular, **TMTQ** is an unusual molecule because the dicyanomethyl groups at both sides cause a unique quinoidal structure, the alternation pattern between C-C and C=C bonds, in the ground state (Figure 4b in the manuscript). Thus, in our analyses, the ground state fragments without one or both of dicyanomethyl groups did not reflect the distinctive quinoidal character of mother system, **TMTQ**. Taking these points into consideration, we analyzed the M10A core dication fragments, where the triplet dicationic character well reflects the quinoidal nature of **TMTQ** system with its aromaticity change (Figure R5). Here, upon structural planarization with enhanced aromatic nature, the C=C stretching IR intensity of M10A dication becomes significantly attenuated, which is well matched with our experimental observations and provides a reliable support for our IR data interpretation. Therefore, we put this information in our revised supplementary information as Figure S13. Once again, we appreciate the reviewer’s considerate advice to improve our manuscript.

Figure R5. The calculated IR spectra of dicationic core annulene in the T_1 state with B3LYP-D3/6-311G(d,p) TD-DFT calculations. The structures **1** and **4** of dicationic core annulene were obtained from the S_0 -state and T_1 -state optimized structures of **TMTQ**, respectively. **2** and **3** are intermediate structures of conformational change in going from **1** to **4**.

>>>(Page 13, line 4 in the revised manuscript)

For more deliberate interpretation for the IR spectral changes, we have also analysed the C=C stretching IR bands of triplet M10A dication fragment in various degrees of structural distortion (Fig. S13), where the quinoidal nature of **TMTQ** system with its aromaticity change was well reflected by the triplet dicationic character. The TD-DFT frequency results showed that, upon the structural planarization with enhanced aromatic nature, the C=C stretching IR intensity of M10A dication becomes significantly attenuated, which is well matched with our experimental observations.

>>>(Figure S13 in the revised Supplementary Information)

Figure S13. The calculated IR spectra of dicationic core annulene in the T_1 state with B3LYP-D3/6-311G(d,p) TD-DFT calculations. The structures **1** and **4** of dicationic core annulene were obtained from the S_0 -state and T_1 -state optimized structures of **TMTQ**, respectively. **2** and **3** are intermediate structures of conformational change in going from **1** to **4**. The triplet dicationic character well reflects the quinoidal nature of **TMTQ** system with its aromaticity change.

Fourth comment from Reviewer 4: *In their response to Referee 3's second query, the authors have provided a Figure showing their spectroelectrochemical setup. The problems of diffusion/spectral purity can be avoided by aiming the FTIR laser through the working electrode grid. The authors seem (based on Figure 3-1) to have aimed their beam through a region of electrolyte, rather than through the electrode. There should be no particular deleterious effect on the FTIR spectra from the presence of the*

grid, but much purer spectra (ie pure oxidation states) will be obtained. It should certainly be possible to measure pure dianion by holding a potential beyond that at which the dianion is generated – measurement of the monoanion may require multicomponent curve resolution, or may be practically impossible if the monoanion spectrum is a linear combination of those for the neutral and dianion states. I remain to be entirely convinced that two electrons transferred in the photophysical experiments, and not just one. Based on the authors' explanation, the ground-state spectrum of 'dianion' in Fig 2g (left, lower) is actually probably monoanion, or at best a mixture of neutral, dianion, and monoanion. It is certainly **not** pure dianion! Any dianion formed at the working electrode would disproportionate by the time it got to the region measured by the FTIR (shown in Fig 3-1). It would be good to repeat the spectroelectrochemical measurements to obtain pure spectra of monoanion and dianion, and use these to assign the TRIR spectrum in Fig 2g (top, left). The most compelling assignment would come from the use of both peak positions and relative intensities/oscillator strengths..

Our response: It is well known that most of the electrochemical reductions of tetracyano π -conjugated compounds show only one two-electron reduction wave in the cyclic voltammetry experiment (*J. Am. Soc. Chem.*, **2002**, *124*, 12380-12388). In addition, in most of these compounds, in the infrared spectroelectrochemical experiment (like that in **TMTQ**), the spectral evolution goes from the neutral directly to the dianion without traces of radical anion. We firmly think this is our case here. No disproportionation reaction occurs in our reduction experiment. The particular setup and configuration of the used thin layer spectroelectrochemical cell, in which we carry out the experiment, definitively helps to avoid the disproportionation reaction. The optical path of the IR beam in the experiment crosses mainly two differentiated parts: 1) the solution which is outside the volume where the reduction takes place (i.e., defined by the Nernst diffusion layer and that consequently gives rise to the spectrum of the neutral molecule; and 2) the Nernst diffusion layer region of the cell in which the electrochemical reaction forms the dianion without intermediate oxidation states. In these conditions, we are confident that the spectra of the reduced species we obtained correspond exclusively to the dianion without traces of the radical anion.

The evidence that we get only the dianion species is that the infrared spectra of radical anion and dianions of similar molecules are well differentiated such as in the case of **TCNQ** (tetracyanoquinodimethane) or **TCNE** (tetracyanoethylene, *Angew. Chem. Int. Ed.*, **2006**, *45*, 2508-2525). So even trace amount of radical anion of **TMTQ** that could be formed would be clearly observed and distinguished.

In addition, we agree with the referee in the fact that if the volume with the neutral compound (outside the Nernst diffusion layer region) and that with the dianion (inside the Nernst diffusion layer region) would mix then a disproportionation reaction would occur. However, this “mixture” does not happen neither in the voltammetry cyclic experiment nor in our spectroelectrochemical experiment.

For the dianion and CT state, it is obvious that the dianion species of **TMTQ** obtained by electrochemical reduction differ from the CT state obtained after photoexcitation in the fact that the CT state bears a dication in the central core and the dianion does not. The electron withdrawal effect of this central dication over the electron density on the external cyano groups (to which the CN stretching frequency is very sensitive) in the CT state would alter the amount of electron transfer or delocalization in this CT state compared with the case of the dianion. However, both frequencies in the CT and electrochemical dianion are the same. The possible difference is cancelled by the radical delocalization in the CN groups (not existing in the dianion) that is favored in the CT state due to the multiconfigurational character.

Reviewers' comments:

Reviewer #2 (Remarks to the Author):

In response to my queries, the authors provided a new set of comments and revisions. This time, I feel that the discussion reached the necessary level of completeness. Even if some assertions may be speculative at this point, the work provides a unified spectroscopic picture of a really interesting molecule, and may be recommended for publication.

Reviewer #4 (Remarks to the Author):

The authors have dealt with most of my concerns, except the final one (relating to spectroelectrochemistry).

In the authors' previous response to referees, they presented a figure showing their spectroelectrochemical cell setup, indicating that the light beam passed outside the working electrode.

This setup is not, in my view, optimal. It is preferred to direct the light source through the Pt grid electrode. The design of a typical OTTE cell is such that the depth of analyte solution on either side of the working electrode (bounded by the cell windows) is much smaller than the thickness of the diffusion layer. In this way, 'pure' spectra can be obtained when the beam passes through the electrode.

In their recent response, the authors state that:

- (a) The beam passes through the 'neutral' region of the cell, giving spectrum of neutral molecules, and...
- (b) the beam passes through the 'reduction' region of the cell (within the electrochemical diffusion layer), giving spectrum of the dianion, but...
- (c) it is impossible for these solutions to mix. If they did mix, then the authors concede that disproportionation would occur

The authors' assertion that this 'mixture' is impossible does not appear to be well founded, but I would appreciate clarification on this point. If they have indeed measured the spectrum using the experimental setup shown in the earlier response to referees (Fig 3.1 in that response), then their beam clearly passed through a region in which reduced species from the working electrode are diffusing into the neutral solution. This diffusion doesn't just 'stop' so some admixture is guaranteed: the concentration profile of reduced species will reduce asymptotically to zero with distance from the electrode; there's no interface as such between reduced and bulk material.

The authors' reference relates to TCNE, a much smaller compound which I would expect to have different electrochemical properties, since the CN groups are much more closely linked. Nonetheless, the paper doesn't really support their argument: the CN stretch frequency which they report for TMTQ²⁻ (2170 cm⁻¹) is in the region reported for the monoanion of TCNE (2150 cm⁻¹ – 2200 cm⁻¹).

It should be possible to get a pure spectrum of dianion using spectroelectrochemistry, with no contamination from 'neutral' species. Alternatively, other methods can be used, such as chemical reduction.

I suggest the following means to conclusively resolve this issue:

1. Re-measure the spectroelectrochemistry of TMTQ, passing the probe beam through the Pt working electrode

2. Generate the dianion of TMTQ chemically, by addition of an excess of suitable reducing agent

I suggest that it is important to have a pure spectrum of the dianion since much of the interpretation relies on having made the dianion by double CT in the TRIR experiment.

It is, of course, possible that the spectrum of the monoanion is the same as a linear combination of the dianion and neutral species, if the M10A linker does not permit strong electronic communication between the CN groups (i.e. Robin-Day Class 1 molecule). Then the monoanion would not have a distinct spectrum.

Minor comment: annulene is misspelled in the caption to Fig 3.

Response to Reviewers' Comments

First of all, we are sincerely appreciative of the reviewers who have obviously taken considerable time out of their own busy lives to help us make this a better paper through their thoughtful consideration and advice. Our responses and specific revisions to the reviewers' comments are shown below.

[Response to Reviewer 4]

The comment from Reviewer 4: *The authors have dealt with most of my concerns, except the final one (relating to spectroelectrochemistry).*

In the authors' previous response to referees, they presented a figure showing their spectroelectrochemical cell setup, indicating that the light beam passed outside the working electrode. This setup is not, in my view, optimal. It is preferred to direct the light source through the Pt grid electrode. The design of a typical OTTLE cell is such that the depth of analyte solution on either side of the working electrode (bounded by the cell windows) is much smaller than the thickness of the diffusion layer. In this way, 'pure' spectra can be obtained when the beam passes through the electrode.

In their recent response, the authors state that:

- (a) The beam passes through the 'neutral' region of the cell, giving spectrum of neutral molecules, and...*
- (b) the beam passes through the 'reduction' region of the cell (within the electrochemical diffusion layer), giving spectrum of the dianion, but...*
- (c) it is impossible for these solutions to mix. If they did mix, then the authors concede that disproportionation would occur*

The authors' assertion that this 'mixture' is impossible does not appear to be well founded, but I would appreciate clarification on this point. If they have indeed measured the spectrum using the experimental setup shown in the earlier response to referees (Fig 3.1 in that response), then their beam clearly passed through a region in which reduced species from the working electrode are diffusing into the neutral solution. This diffusion doesn't just 'stop' so some admixture is guaranteed: the concentration profile of reduced species will reduce asymptotically to zero with distance from the electrode; there's no interface

as such between reduced and bulk material.

The authors' reference relates to TCNE, a much smaller compound which I would expect to have different electrochemical properties, since the CN groups are much more closely linked. Nonetheless, the paper doesn't really support their argument: the CN stretch frequency which they report for TMTQ^{2-} (2170 cm^{-1}) is in the region reported for the monoanion of TCNE ($2150\text{ cm}^{-1} - 2200\text{ cm}^{-1}$).

It should be possible to get a pure spectrum of dianion using spectroelectrochemistry, with no contamination from 'neutral' species. Alternatively, other methods can be used, such as chemical reduction.

I suggest the following means to conclusively resolve this issue:

1. Re-measure the spectroelectrochemistry of TMTQ, passing the probe beam through the Pt working electrode
2. Generate the dianion of TMTQ chemically, by addition of an excess of suitable reducing agent

I suggest that it is important to have a pure spectrum of the dianion since much of the interpretation relies on having made the dianion by double CT in the TRIR experiment.

It is, of course, possible that the spectrum of the monoanion is the same as a linear combination of the dianion and neutral species, if the M10A linker does not permit strong electronic communication between the CN groups (i.e. Robin-Day Class 1 molecule). Then the monoanion would not have a distinct spectrum.

Our response: We are glad that the reviewer recognized our efforts to deliver clearer description of our scientific findings. And we also sincerely appreciate the reviewer's supportive comment and considerate advice for letting our manuscript more reliable and obvious. As the reviewer mentioned, we also agree with the importance of IR spectra of **TMTQ** dianion because its $\text{C}\equiv\text{N}$ stretching IR band plays a critical role in our scientific discovery, 'Two-Electron Transfer Stabilized by Excited-State Aromatization'. In this regard, according to the reviewer's kind advice, we have carefully remeasured the IR spectrum of electrochemically reduced **TMTQ** in the region of $2140\sim 2240\text{ cm}^{-1}$, for the $\text{C}\equiv\text{N}$ stretching IR bands. To solve the residual problem, we have reduced the length of the path traversed by the IR beam. With this, there is a lack of absorbance but also we are able to reduce the volume of the solution volume outside the diffusion layer, and obtained the clear IR spectrum manifesting the pure **TMTQ** dianion obviously.

Figure 1 below shows that recording the IR spectra under application of an overpotential of $\approx -0.8\text{ V}$, we achieved almost full disappearance of the $\text{C}\equiv\text{N}$ stretching IR band from neutral **TMTQ** which takes place in parallel with the progressive emergence of the band at 2170 cm^{-1} of **TMTQ** dianion. There is a

clear isosbestic point between the spectra of the neutral and dianion species which reveals that the conversion goes through an equilibrium of these two species converting one into the other. Thus, the blue spectrum in Figure 1b is now clearly due to a >90-95% of the conversion of neutral **TMTQ** into the dianion thus corresponding to its almost pure IR spectrum.

Figure 1. (a) The IR spectroelectrochemistry in the reduction wave of **TMTQ** at -0.8 V overpotential. (b) The cyclic voltammogram of **TMTQ**. (c) The UV-Vis-NIR absorption spectrum of **TMTQ** recorded during the electrochemical reduction. The red-color and blue-color lines indicate the initial and final species during the spectroelectrochemistry, respectively.

To clarify more this situation we also display in Figure 1c, the UV-Vis-NIR absorption spectra of **TMTQ** upon the electrochemical reduction carried out in the same conditions as the IR spectroelectrochemistry of Figure 1a. Again full conversion of the neutral into the dianion with a clear isosbestic point is observed with a main band at 560 nm for the dianion.

We have also scanned the UV-Vis-NIR spectroelectrochemical response of **TMTQ** in the oxidation part and this is shown in Figure 2. During oxidation, the neutral species converts into an oxidized species which is characterized by a band around 470 nm (Figure 2a). Interestingly, the position of this band concurs with the PIA (photoinduced absorption) band in the transient absorption (TA) spectra, which was assigned arising from the two-electron CT state (see red arrows in Figure 2a and 2b). This suggests that this transient species have a core with electron deficient character which justifies the resemblance of the bands between the TA spectra and UV-Vis-NIR spectra of the oxidized form of **TMTQ**.

From these spectroscopic comparisons, the CT state of **TMTQ** can be clearly described with the shift of two π -electrons from the central M10A donor to both DT acceptors. To deliver this information, we change Figure 2g and put additional figures, Figure S7, in the supplementary information.

Figure 2. (a) The UV-Vis-NIR absorption and (b) transient absorption spectra of **TMTQ**. The UV-Vis-NIR absorption spectrum of **TMTQ** recorded with spectroelectrochemistry in the oxidation wave up to +1.0 V, where the spectra of initial and oxidized species are colored as red and blue, respectively.

>>>(Figure S7 in the revised Supplementary Information)

Figure S7. (a) The FT-IR spectra of neutral (red line) and electrochemically produced dianion (blue line) **TMTQ** in the region of 2140–2240 cm^{-1} for the C≡N stretching vibrational modes.

>>>(Figure 2 in the revised manuscript)

Fig. 2 | Electronic and vibrational spectra of TMTQ. a-d. The steady-state (a) and transient absorption spectra of TMTQ in toluene (b), CH₂Cl₂ (c), and CH₃NO₂ (d). The inset plots are the decay profiles at 660 nm. e,f. The FT-IR (e) and transient IR (f) spectra of TMTQ in toluene (top), CH₂Cl₂ (middle), and CH₃NO₂ (bottom). The transient IR spectra were plotted within ~100 ps time window (from purple to dark red colored lines). g. The transient IR spectra of TMTQ in CH₂Cl₂. The spectra in the regions of 2140–2240 and 1275–1575 cm⁻¹ are assigned to IR bands for C≡N and C=C stretching vibrations, respectively. The FT-IR spectra of neutral and electrochemically produced dianion TMTQ are inset.

REVIEWERS' COMMENTS:

Reviewer #4 (Remarks to the Author):

Thank you for completing the extra SEC experiment. The data support the generation of dianion in the excited state and so my only remaining concern is resolved.

Congratulations on the very nice work. I am happy to recommend it for publication.